# The Close Relationship between the Golgi Trafficking Machinery and Protein Glycosylation

**DOI:** 10.3390/cells9122652

**Published:** 2020-12-10

**Authors:** Anna Frappaolo, Angela Karimpour-Ghahnavieh, Stefano Sechi, Maria Grazia Giansanti

**Affiliations:** Istituto di Biologia e Patologia Molecolari del CNR, Dipartimento di Biologia e Biotecnologie, Sapienza Università di Roma, 00185 Roma, Italy; angela.karimpourghahnavieh@uniroma1.it (A.K.-G.); stefano.sechi@cnr.it (S.S.)

**Keywords:** glycosylation, Golgi, GOLPH3, GORAB, oligomeric golgi complex

## Abstract

Glycosylation is the most common post-translational modification of proteins; it mediates their correct folding and stability, as well as their transport through the secretory transport. Changes in *N-* and *O-*linked glycans have been associated with multiple pathological conditions including congenital disorders of glycosylation, inflammatory diseases and cancer. Glycoprotein glycosylation at the Golgi involves the coordinated action of hundreds of glycosyltransferases and glycosidases, which are maintained at the correct location through retrograde vesicle trafficking between Golgi cisternae. In this review, we describe the molecular machinery involved in vesicle trafficking and tethering at the Golgi apparatus and the effects of mutations in the context of glycan biosynthesis and human diseases.

## 1. Protein Glycosylation and the Golgi

The enzymatic addition of glycans to proteins, dubbed “protein glycosylation”, is a fundamental post-translation and co-translational modification that controls the folding and stability of proteins and their transport through the secretory pathway [1,2,3]. The composite repertoire of glycan structures in glycoproteins impacts on many different cellular and developmental processes such as signal transduction, molecular trafficking, cell-cell communication, immunity and early mammalian development [4,5]. Since one fifth of all proteins in structural databases are classified as glycoproteins, it is not surprising that up to 2% of the eukaryotic proteome is required for glycosylation [3,6]. It has been estimated that 700 proteins are required to generate the glycan structures in human glycome [7]. Glycoprotein glycosylation is a sequential process that involves the coordinated action of hundreds of glycosyltransferases (GTs) and glycosydases, which are trafficked to the specific locations of the endoplasmic reticulum (ER) and Golgi apparatus [8]. Only ten monosaccharides are essential for the synthesis of animal glycans: glucose (Glc), galactose (Gal), *N*-acetylglucosamine (GlcNAc), *N*-acetylgalactosamine (GalNAc), fucose (Fuc), mannose (Man), xylose (Xyl), glucuronic acid (GlcA), sialic acid (SA, either as *N*-acetylneuraminic acid, Neu5Ac or 5-*N*-glycolylneuraminic acid acid, Neu5Gc). The main types of protein glycoprotein glycans are *N*- or *O*-linked. *N*-glycosylation starts in the ER with the synthesis of a glycan precursor with the composition Glc3Man9GlcNac2 on the carrier dolichol isoprenoid lipid [9]. The precursor glycan is then transferred en bloc onto asparagine residues of nascent proteins by multisubunit oligosaccharyltransferase (OST) complexes within the lumen of the ER [10,11,12]. Distal glucose moieties of the Glc3Man9GlcNac2 glycan are trimmed before reaching the Golgi apparatus by sequential actions of ER glucosidases, representing an important step in the control of folding processes for secretory pathway glycoproteins prior to ER exit [13,14,15]. Glycoproteins that exit the ER reach the Golgi apparatus with eight or nine mannose residues for further processing of the glycan moieties into complex and hybrid glycan forms (Figure 1). 

The Golgi apparatus is organized into discrete cisternae, each containing a distinct subset of glycosylation machinery proteins which include glycosyltransferases, glycosidases and nucleotide sugar transporters [16,17]. As the secretory proteins arrive at the *cis*-Golgi, mannose trimming generates the Man5GlcNAc2Asn intermediate followed by the transfer of GlcNAc residue to a terminal Man residue on the α3-arm of the glycan structure by the medial Golgi GT GlcNAcT-1. The addition of GlcNAc to Man5GlcNAc2Asn is an essential step to generate hybrid and complex *N*-linked glycans [15,17]. Hybrid *N*-glycans are generated by the extension of the α3-arm that received GlcNAc with the addition of Gal, GalNAc, Fuc and SA. Removal of two terminal of the five Man residues, allows branching with an additional GlcNAc residue and synthesis of biantennary complex *N*-glycans. Further branching may occur, leading to multiantennary complex glycans in *medial-late* and *trans*-Golgi compartments [17,18]. Complex glycans can be decorated with Gal, GalNAc, Fuc and SA. 

In contrast to *N*-linked glycosylation, *O-*glycosylation does not involve a lipid-linked oligosaccharide precursor for transfer to nascent polypeptides. Instead, it consists of the formation of a single glycosidic linkage between Ser or Thr residues and GalNAc, Man, Glc, Xyl, or Fuc [17]. The most abundant type of *O-*glycans are the mucins initiated by GalNAc-Ser/Thr and the glycosaminoglycan (GAG) chains on proteoglycans initiated by a conserved tetrasaccharide Glc-A-β1,3-Gal-β1.3-Gal-β1,4Xyl-β (Figure 1). Synthesis of *O-*glycans starts in the early *cis*-Golgi or in a translational compartment (ERGIC, with characteristics of the ER), and is completed via the sequential addition of glycans while the secretory protein moves through the *cis-*, *medial-* and *trans-*Golgi cisternae. 

Efficient protein *N*- and *O*-linked glycosylation is strictly dependent on the precise distribution of glycosyltransferases, glycosidases and nucleotide sugar transporters which should be maintained in the correct Golgi compartment as the secretory cargo protein passes through the Golgi. How Golgi enzymes are retained in the Golgi membranes is strictly regulated by vesicle trafficking across the Golgi [19]. 

So far, more than 100 monogenic congenital disorders of glycosylation (CDGs) have been identified and associated with a wide variety of symptoms such as neurological deficit, liver disfunction, and skin and bone alterations [20,21]. Traditionally CDGs have been divided into two groups: CDG-I affect the synthesis and/or transfer of the dolichol pyrophosphate oligosaccharide precursor of *N*-linked glycoproteins, while CDG-II affect the processing of *N*-linked glycans or the biosynthesis of *O-*linked glycans [22]. Many pathogenic mutations in intra-Golgi trafficking regulators have been associated with CDG II. This review will highlight the strict relationship between the Golgi trafficking machinery and protein glycosylation.

## 2. Enzyme Sorting at the Golgi Cisternae

Several models for Golgi trafficking have been postulated, but the most favored model for membrane traffic within the Golgi is the cisternal maturation model of Golgi transport ([19,23]; Figure 2). 

According to the cisternal maturation model, as a new *cis*-Golgi cisterna is formed, it is gradually converted by the accumulation of *medial* and then *trans* glycosylation enzymes that travel from more mature Golgi compartments in vesicles moving in retrograde direction. Although there is strong support for this model, recent observations of the transport of soluble cargoes in mammalian cells have led researchers to postulate the existence of intercisternal tubules that offer specialized Golgi traffic routes [24]. Retrograde transport and sorting of glycosylation enzymes depends mostly on COPI-coated vesicles, composed of seven different protein subunits αCOP, βCOP, β’COP, γCOP, δCOP, εCOP and ζCOP [25]. The COPI coat is recruited from the cytosol to the membrane primarily through binding to activated Arf1 GTPase. As coat proteins are recruited, they assemble a cage-like structure that enhances membrane curvature to promote vesicle formation [26,27]. A recent study showed that COPI carriers can traffic in anterograde as well as retrograde directions, with the small GTPase Cdc42 regulating the bidirectional COPI transport specificity [28]. Golgi glycosylation also requires maintenance of a stringent cisternal pH, which influences retrograde sorting of enzymes and the association of a glycosylation enzyme with a trafficking cargo [29]. Indeed, disruption of the cisternal pH leads to mislocalization of glycosyltransferases [30,31]. Consistent with the role of the pH in Golgi glycosylation, loss-of-function mutations in *ATP6V0A2*, the gene encoding the membrane-bound V0a2 subunit of the V-ATPase, disrupt retrograde intra-Golgi trafficking, resulting in glycosylation defects [32]. Patients carrying pathogenic variants in *ATP6V0A2* present with wrinkly skin syndrome, autosomal recessive cutis laxa type II and neurological impairment [32]. Recent work showed that Golgi localization of glycosylation enzymes and their function also depend on lipid composition of the Golgi compartments [33,34]. Disruption of sphingomyelin homeostasis at the *trans*-Golgi network of mammalian cells, by using short-chain ceramide, affects the organization of Golgi membranes, leading to separation of Golgi-resident proteins from each other [33]. As a consequence, the enzyme sialyltransferase is unable to interact with its substrate TGN46, that subsequently fails to be glycosylated. Moreover, the biosynthesis of complex sphingolipids is required for the retention of a Golgi mannosyltransferase in yeast [34].

## 3. Molecular Actors in Vesicle Targeting and Fusion at the Golgi

The fusion of transported COPI vesicles at the target Golgi membrane requires the orchestrated action of Rab GTPases, tethering complexes and SNARE proteins [20,35,36,37]. Small Rab GTPases are important molecular regulators of multiple vesicle trafficking steps [38]. There are about 20 different Rab proteins that associate with the Golgi membranes [36]. In the GTP-bound active state (membrane-associated), Rab proteins recruit molecular motors and tethering factors to specific membrane domains to promote vesicle fusion [39]. Prior to fusion, transport vesicles and their cognate target membranes must be properly aligned, a process known as tethering. The tethering of vesicles at the cognate target membrane is regulated by large coiled-coil tethers (CCTs, [40,41,42,43,44]) or multisubunit tethering complexes (MTC; [35,45]). CCTs that localize at the Golgi apparatus are named ˝golgins˝ [43]. In humans, there are 11 golgins including GM130, Golgin-45, Golgin-97. Golgins are anchored to the Golgi membranes by the carboxy-terminus, and protrude into the cytoplasm to capture vesicles, usually through the amino-terminus of the protein [40]. These tethers are large dimeric proteins displaying two globular heads connected by a long coiled-coil structure, which greatly facilitate the first contact with the vesicles over distances of more than 200 nm [35]. Golgins can occupy several locations within the Golgi, residing at the *cis*-Golgi, the rims of the Golgi stacks and the *trans*-Golgi network. Due to their ability to selectively tether vesicles of specific cargo content, these proteins play an essential role in directing vesicle trafficking within the Golgi. Most golgins have been found to cooperate with several trafficking players including Rab proteins and other GTPases, MTCs and SNAREs [46]. Rabs bind the coiled-coil regions within the golgin protein, whereas small Arf GTPases interact with GRAB/GRIP domains, located in the terminus of many golgins [36].

The multisubunit tethering complexes (MTCs), consisting of three to ten subunits and an overall molecular weight of 250–800 kDa, can associate with vesicles over much shorter distances compared to CCTs [45]. Recent structural studies and sequence comparisons have led researchers to group MTCs into CATCHRs and non-CATCHRs [47]. The family of complexes associated with tethering containing helical rods (CATCHR) include DSL1/ZW10 complex, Conserved Oligomeric Golgi (COG), GARP (Golgi associated retrograde complex) and the exocyst [45]. The non-CATCHR group includes two complexes involved in regulating vesicle transport from late Golgi to endosomal/lysosomal compartments named Homotypic fusion and protein sorting (HOPS) and class C core vacuole/endosome tethering (CORVET) [45]. The multisubunit Transport Particle (TRAPP) complexes (I, II and III) have been classified as MTCs with a unique structure [48]. TRAPPI function is involved in ER-Golgi transport [49]. 

Although MTCs and CCTs function as tethers, they are implicated in different steps of vesicular tethering [50]. The long CCTs are required for the initial capturing of distant vesicles through transient and low affinity interactions. Conversely MTCs, which can bind vesicles at short distances from the target membrane, act in later events of tethering and facilitate SNAREpin in assembly. Thanks to their multiple subunit structures, MTCs can simultaneously interact with various vesicle trafficking regulators (i.e., CCTs, Rab1 and SNAREs), playing a key role in coordinating vesicular tethering with docking and fusion [35].

The fusion of the uncoated vesicle with the target membrane at the Golgi requires the assembly of functional SNAREs (soluble *N*-ethylmaleimide-sensitive factor activating protein receptors) complexes [50]. SNARE proteins share a characteristic coiled-coil motif of approximately 70 amino acids comprising heptad repeats, and can localize on either the target (t-SNARE) or the vesicle membrane (v-SNARE). The interaction of cognate v-SNAREs and t-SNAREs leads to the formation of a fusogenic *trans*-SNARE complex or SNAREpin, in which four SNARE motifs form a twisted parallel four helix-bundle, bringing the membranes together and leading to vesicle fusion [51,52]. Sec1/Munc18 (SM) proteins, evolutionarily conserved peripheral membrane proteins of 60–90 kDa, act in conjunction with tethering factors and SNAREs [50,53]. SM proteins can bind to the syntaxin family of SNAREs [54] or to the whole SNARE complex [55,56,57]. In both cases SM proteins have been shown to promote SNAREpin assembly and membrane fusion at the Golgi membranes.

## 4. GOLPH3 Interacts with COPI to Recruit Glycosyltransferases to Golgi Compartments

Golgi phosphoprotein 3 (GOLPH3), a highly conserved protein from yeast to humans, has been characterized as a Phosphatidylinositol 4-phosphate [PI(4)P] effector that regulates Golgi trafficking [58]. Identified in proteomic-based studies of the Golgi [59,60], GOLPH3 localizes to the *trans*-Golgi via the direct interaction with PI(4)P, mediated by its unique C-terminal GPP34 domain [61,62]. Localization of GOLPH3 protein to the Golgi is required to maintain Golgi architecture in human cells and *Drosophila melanogaster* [61,63]. 

Much evidence indicates that GOLPH3 is an oncogene, that is overexpressed in several solid tumors such as lung cancer, breast cancer, colon cancer and melanoma [58,64]. However, the molecular mechanisms underlying the oncogenic properties of GOLPH3 are poorly understood. It has been proposed that GOLPH3 might contribute to cellular transformation by affecting the glycosylation of key cancer relevant glycoproteins [58]. Experimental data from human cells and in model organisms indicate that GOLPH3 plays a fundamental role for retrograde intra-Golgi trafficking of protein glycosyltransferases ([58]; Figure 3). 

Glycosyltransferases are all type II integral membrane proteins containing of a short cytosolically exposed *N*-terminal region, a single membrane-spanning domain and a luminal enzymatic domain [5]. The cytoplasmic tails of Golgi glycosyltransferases lack the canonical COPI-binding sorting signals. The role of GOLPH3 in enzyme recycling was initially demonstrated for the yeast homologue Vps74p which was shown to bind the *N*-terminal tails of a set of mannosyltransferases as well as the COPI coatomer, thereby anchoring these enzymes at the Golgi [67,68]. Mutations in the PI(4)P-binding pocket of Vps74p caused loss of Vps74p-membrane association in yeast cells and mislocalization of mannosyltransferases [69]. Both GOLPH3 and Vps74p form oligomers and tetramerization of Vps74p was essential to bind to a pentameric sequence motif at the cytoplasmic tail of a subset of Golgi glycosyltransferases in *in vitro* assays [70]. A cluster of evolutionarily conserved arginine residues at the *N*-terminus of GOLPH3 proteins was shown to mediate coatomer binding [70]. Vps74p-coatomer binding was required for retention of glycosyltransferases to the Golgi, but not for Vps74p association with the Golgi membrane which was instead dependent on the oligomerization status and the PI(4)P binding capabilities [70]. Vps74p recognizes a (F/L)(L/I/V)XX(R/K) motif in the cytoplasmic tail of numerous yeast mannosyltransferases raising the possibility that it might function as an adaptor for incorporation of glycosyltransferases in the COPI vesicles. A sequence similar to the (F/L)(L/I/V)XX(R/K) motif, was also found in the cytoplasmic tail (CT) of Core 2 *N*-acetylglucosaminyltransferase 1 (C2GnT1; [71]). C2GnT1 plays a crucial role in the synthesis of core 2-associated sialyl Lewis x (C2-O-sLex), a ligand involved in selectin-mediated leukocyte trafficking and tumorigenesis [72]. GOLPH3 protein interacts with the CT of C2GnT1 via a LLRRR sequence and controls Golgi localization of C2GnT1 in KG1a and K562 cells [71]. Moreover, depletion of GOLPH3 affects synthesis of C2-O-sLex associated with P-selectin glycoprotein ligand-1 with effects on cell tethering, rolling and adhesion [71].

The involvement of GOLPH3 in *O-*glycosylation was further demonstrated by its interaction with the CT of POMGnT1, the *O*-mannosyl-β-1,2-*N*- acetylglucosaminyltransferase 1 required for *O-*mannosylation of α-dystroglycan [73]. α-dystroglycan, a key component of the α-dystroglycan glycoprotein complex mediates the association with laminin and other components of the extracellular matrix [74]. Perturbation of the glycosylation status of α-dystroglycan has been linked to various forms of congenital muscular dystrophies [75]. Loss of GOLPH3/ POMGnT1 interaction impairs Golgi localization of POMGnT1 and results in a reduction in IIH6 immunoreactivity indicating a defective glycosylation status of α-dystroglycan.

Chang and coauthors [76] showed that GOLPH3 controls the ability of Golgi to retain exostosins, a class of glycosyltransferases implicated in *O-*glycosylation of heparan sulfate proteoglycans. Exostosins catalyze the addition of alternating β1-4-linked glucuronic acid (GlcA) and α1-4-linked *N*-acetylglucosamine (GlcNAc) disaccharides in the biosynthesis of glycosaminoglycan chains [77]. In humans, mutations affecting the exosostins EXT1 and EXT2 cause multiple osteochondrosoma (MO), an autosomal dominant skeletal disease characterized by multiple cartilaginous tumors [78,79,80]. *Drosophila* GOLPH3 physically interacts with the exosostin EXT1and EXT2. Moreover, knockdown or overexpression of GOLPH3 affects the distribution of EX1 and EX2 proteins within the Golgi cisternae, resulting in defective HSPGs and Hedgehog signaling [76]. Consistent with the results in *Drosophila*, Golgi localization of EXT1 and EXT2 is sensitive to GOLPH3 protein levels in cells cultivated from bone cells such as osteosarcoma cells (U2OS, MG63), chondrosarcoma cells (SW1353), and rhabdomyosarcoma (RD) cells [76]. 

Several studies have supported the role of the GOLPH3 protein in *N*-glycosylation ([65,81]; Figure 3). Isaji and coworkers [81] showed that GOLPH3 knockdown (KD) affects integrin-mediated cell migration which correlates with defective sialylation of *N*-glycans, especially α2,6-sialylation on β1-integrin. They further demonstrated that GOLPH3 associates with α2,6-sialyltransferase-I (ST6GAL1) in HeLa cells and that expression of α2,6-sialyltransferase-I (ST6GAL1) rescues integrin-dependent cell migration and cellular signaling defects in GOLPH3 KD cells. Eckert and coauthors [65] reported a role of human GOLPH3 in spatial regulation of ST6GAL1. They showed that GOLPH3 protein directly interacts with Core 2*N*-acetylglucosaminyltransferase (C2GnT) and ST6GAL1, and directs incorporation of these enzymes into COPI coated vesicles. Conversely, galactosyltransferase, an enzyme that does not bind to GOLPH3, is not incorporated into COPI vesicles, and does not depend on GOLPH3 for its Golgi localization. Remarkably, previous studies reported that increased α2-6 sialylation of β1-integrins affect the adhesive and migratory capacity of tumor cells, and may contribute to colon tumor progression [82,83]. Collectively, these data suggest a strong correlation between GOLPH3-dependent glycosylation defects and tumor progression.

## 5. GORAB, a Scaffolding Protein for COPI, Is Involved in Gerodermia Osteodysplastica

The RAB6-interacting protein GORAB, also known as NTKL-binding protein 1, was also dubbed “Scyl1-binding protein 1” after its interaction with Scyl1/NTKL protein [84]. Scyl1, a component of the Scy1-like family of catalytically inactive protein kinases, was involved in COPI-mediated retrograde trafficking because of its association with isoform 2 of γCOP subunit, through a C-terminal RKXX-COO- dibasic motif [85,86]. *GORAB* loss-of-function mutations have been linked to Gerodermia osteodysplastica GO, OMIM 231070; [87], an autosomal recessive inherited disorder characterized by lax and wrinkly skin, which gives patients a prematurely aged appearance [87,88]. Moreover, individuals suffering from GO exhibit a severe osteoporosis with reduced bone mass, susceptibility to fractures and malar and mandibular hypoplasia [87,88]. To date, the molecular mechanisms that link loss of GORAB with the onset of GO disease remain to be clarified. Evidence indicates that GORAB is a key regulator of the E3 ligase Mdm2, promoting both its transcription, by binding *Mdm2* promoter [89], and Mdm2 protein self-ubiquitylation [90]. A recent work implicated the *Drosophila* homologue of GORAB in centriole structure and duplication [91]. Although the above studies proved a subcellular localization of GORAB in the cytoplasm and in the nucleus, the GO disease has been mostly associated with Golgi-associated functions [87,92]. Indeed, in dermal fibroblasts and osteoblasts, which represent the mostly affected cell types in GO, GORAB protein was detected almost exclusively at the Golgi apparatus [87,92]. 

Originally classified as a member of the Golgin family, GORAB localizes to the *trans*-Golgi through the association with the active, GTP-bound, form of the small RAB6 and ARF5 GTPases through the “internal Golgi-targeting RAB6 and ARF5 binding domain” (IGRAB) [87,92]. However, a recent study has suggested that GORAB functions as a scaffolding protein for COPI assembly at the *trans*-Golgi, rather than behaving as a member of the Golgin family [66]. Witkos and coauthors [66] demonstrated that GORAB undergoes oligomerization and localizes to discrete puncta throughout the *trans*-Golgi where Scyl1 and GORAB directly interact via their respective *N*-terminal domains. Moreover, a mutational analysis indicated that the ability of GORAB to self-associate in oligomers is involved in the assembly of the coatomer. Taken together, these data indicate that the GORAB/Scyl1 complex is necessary and sufficient to recruit the COPI complex to the Golgi membranes [66]. GORAB and Scyl1 were also shown to directly bind to the active GTP-bound form Arf1, indicating that they might function as Arf1 effector proteins. Of note, the predominant Arf1 binding coat proteins at the *trans*-Golgi network (TGN) are AP1 and the GGAs [93]. Witkos and coauthors propose that GORAB and Scyl1 oligomers provide discrete domains at the TGN, where high levels of Scyl1 and GTP loaded Arf1 favor the selective COPI coat assembly at the expense of AP1 ([66]; Figure 3). 

The roles of GORAB protein in the COPI-mediated retrograde trafficking also involve maintenance of the proper set of glycosyltransferases in the Golgi *trans*-cisterna. Indeed, in HeLa cells depleted of GORAB, the β-galactoside α-2,6-sialyltransferase 1(ST6GAL1) undergoes a shift towards the later compartments of Golgi [66]. Moreover, *N*-glycome analysis by mass spectrometry from fibroblasts of GO patients and from skin samples of GORAB-knockout mice revealed a significant decrease of sialylated complex glycans [66]. Recent work indicated the requirement for GORAB protein for *O-*glycosylation. An analysis of a Gorab null full knock out in a mouse model revealed a substantial reduction of the dermatan sulfate glycosaminoglycan (GAG) attached to proteoglycans decorin and biglycan through *O-*glycosylation [94]. According to these data, GO can be considered as a type II congenital disorder of glycosylation (CDGs-II), affecting the tissues that are mostly dependent on protein glycosylation and glycanation, such as skin and bone. Indeed, these tissues present a large component of extracellular matrix; for this reason, they could be very sensitive to GORAB mutations [66,87,94].

## 6. The Vesicle Tethering COG Complex and Other Tethering Factors Are Required for Normal Glycosylation

Recent work involved mutations in CCTs in glycosylation defects and CDGs. GMAP-210 functions as a tether for both ER-to-Golgi and intra-Golgi vesicles [95]. Loss-of-function mutations in the gene encoding GMAP-210 are associated with neonatal lethal skeletal dysplasia in mice and achondrogenesis type 1A in humans [96]. Mice carrying null mutations in *Trip11*, which encodes GMAP-210, displayed swollen ER and alterations of Golgi architecture in multiple tissues including cartilage [96]. Glycan processing defects in the Golgi were found in fibroblasts and chondrocytes of mice lacking GMAP-210, resulting in intracellular accumulation of perlecan, but not of type II collagen or aggrecan in chondrocytes. Similarly, fibroblasts from patients carrying either a heterozygous or homozygous nonsense mutant variant of GMAP-210 displayed incomplete glycosylation of the model secretory protein vesicular stomatitis virus G protein, indicating a link between altered glycan processing and GMAP-201 mediated vesicle tethering [96]. 

The requirement for the conserved oligomeric Golgi (COG) complex for normal Golgi-glycosylation has been demonstrated by a large amount of research data [20]. The eight-subunit COG complex, a member of the CATCHR family of tethers, regulates retrograde intra-Golgi vesicle trafficking [97,98,99,100]. The COG complex consists of eight subunits and can exist as a hetero-octameric complex or two distinct subcomplexes: Lobe A (COG1-4) and Lobe B (COG5-8) [101,102,103]. COGI-COG8 interaction, through the formation of alpha-helical bundles, is required to bridge the two lobes together [101,102,104]. Lobe A predominantly associates with the Golgi membranes of every cisterna whereas Lobe B localizes to COPI vesicles [105]. Depletion of COG subunits in yeast and mammalian cells results in a massive accumulation of Golgi-derived vesicles of approximately 60 nm [106,107,108]. These vesicles called COG complex dependent (CCD) vesicles, are enriched in *medial/trans*-Golgi enzymes and intra-Golgi v-SNAREs and can be tethered in vitro by purified COG, demonstrating the role of COG as a tethering complex [109]. The COG complex has a central role in intra-Golgi trafficking due to its ability to associate with several vesicle trafficking players. Indeed, COG subunits were shown to interact with intra-Golgi SNAREs, SM proteins, COPI coat proteins, small Rab GTPases and CCTs [50,110,111]. Several studies indicate that the COG complex may coordinate tethering and vesicle fusion events [50]. Importantly the interaction of the COG complex subunit COG4 with SM proteins and SNAREs is required for the assembly of two different SNAREpin complexes, the intra-Golgi Stx5-GS28-Ykt6-GS15 and the *trans*-Golgi STX16/STX6/VT1A/VAMP4 [112,113]. 

In humans, mutations in *COG1*, *COG2*, *COG4-COG8* cause inherited, autosomal recessive, congenital disorders of glycosylation CDGs-II, [114,115,116,117,118,119,120,121,122,123,124,125]. Deficiencies in COG complex subunits (COG-CDG) have been linked to developmental impairments including microcephaly, mental retardation, hypothonia accompanied with dysmorphic features and feeding problems [114,115,116,117,118,119,120,121,122,123,124,125]. Mutations in *COG7* gene, cause the highest mortality within the first year of life [123,124]. Glycan analysis by mass-spectrometry revealed hyposialylation of serum proteins, abnormal synthesis of *N-* and *O-*linked glycans and altered glycolipids in most COG-CDG patients [114,119,120,122,123,124,125]. COG complex is evolutionarily conserved with COG subunits present in all the eukaryotic kingdom [126,127]. The requirement for COG complex for proper glycosylation was demonstrated in model organisms and humans [106,109,128,129,130,131,132,133,134]. Several groups studied the effects of siRNA-mediated knockdown (KD) of individual COG subunits in HeLa cells in studies that complemented the analysis of patients’ fibroblasts [106,111,133,135,136]. In response to depletion of COG complex subunits, the entire COG complex is non-functional and several glycosylation enzymes including *medial*- (MGAT1 and MAN2A1), and *trans*-Golgi (B4GALT1 and ST6GAL1)- glycosyltransferases were severely mislocalized [106,133]. *N*-glycan profiling revealed decreased levels of sialylation in COG3 and COG4 KDs [133]. Another study reported defects of terminal sialylation in COG3 and COG7 KD [136]. More recently the CRISPR/Cas9 approach was used to generate HEK293T knockout cell lines missing individual COG subunits. KO cell lines displayed retrograde trafficking defects accompanied by sialylation and fucosylation alterations, albeit with effects that varied with the affected subunit [110].

The effects of COG mutations have been studied in many model organisms including yeast, worms and fruit flies, and several phenotypic effects at the cellular and organismal levels have been reported [103]. The loss of COG1 (*cog1*/LDLB cells) or COG2 (*cog2*/LDLC cells) in Chinese hamster ovary (CHO) cells disrupts Golgi structure, resulting in defects of protein sorting and secretion and immature of *N*-, *O*-, and glycolipid glycosylation with reduced sialic acid in glycans [97,106,128,137,138]. Glycosylation defects attributed to Golgi trafficking also characterized the phenotypes of COG mutants in *S. cerevisiae* at the restrictive temperature [130,139,140,141,142]. In *C. elegans* COG1 and COG3 proteins are required for normal glycosylation and gonadal localization of MIG-17 protease, a member of the ADAM family proteins, which controls migration of gonadal distal tip cells [143]. Structural analysis of *N*-glycans by MS showed that *cog1* mutant failed to form the fucose-rich *N*-glycans and to add fucose to residues on *N*-glycans. MS analysis also revealed an increase in high mannose and paucimmanose glycans in *C. elegans cog1* mutants [129]. The *Drosophila* homologues of human COG5, and COG7 are essential for male fertility, normal cytokinesis and spermatid differentiation [144,145,146]. Loss-of-function *Cog7* mutations reduce life span and cause psychomotor defects in *Drosophila* similar to *COG7-CDG* patients [134]. Analysis of *N*-linked glycoprotein glycans by multidimensional ion trap mass spectrometry in heads of *Drosophila Cog7* mutants, revealed altered N-linked glycome profiles with an increased abundance of high mannose and paucimannose type glycans and of a family of neural-specific, difucosylated *N*-glycans. Additionally, *Cog7* mutants displayed hyposialylation. Thus, the phenotypic traits of the *Drosophila* COG7-CDG disease model parallel the pathological characteristics of COG7-CDG patients, including neuromotor defects and defective *N*-glycome profiles. Additionally, *Drosophila Cog7* mutants revealed defects in the larval neuromuscular junctions leading to a reduction in bouton numbers. Different defects in larval NMJs were observed in *Drosophila Cog1* null mutants, which increased synaptic branch length without affecting bouton density [147]. Comparison of *N*-glycome alterations in *Cog1* null mutants with those of *Cog7* mutants should elucidate on the requirements for specific glycans in the larval NMJ architecture. The analysis of COG complex interactome in *Drosophila melanogaster* confirmed the interaction with Rab1 and allowed to identify GOLPH3 as a novel molecular partner of Cog5 and Cog7 [134,148]. Cog 7 protein is required for Golgi localization of GOLPH3 and facilitates GOLPH3 binding with the coatomer and Rab1 GTPase. These results suggest that the COG complex, besides its role in vesicle tethering, might act together with GOLPH3 to coordinate the incorporation of glycosyltransferases into COPI coated vesicles. Overexpression of Rab1 can rescue the locomotor defects caused by loss of *Drosophila* Cog7. Taken together, these results suggest that the *Drosophila* COG7-CDG model can be used to test novel potential therapeutic strategies by modulating trafficking pathways.

## 7. Concluding Remarks

Glycan processing at the Golgi plays an essential role in cellular physiology and differentiation. Defects of *N*-and *O-*linked glycans have been linked to multiple human diseases including CDGs, inflammatory diseases and cancer [149]. Cancer cells often display characteristic glycosylation patterns such as abnormal core fucosylation, high mannose *N*-linked glycans and truncated *O-*glycans Tn (GalNAcα1-Ser/Thr) and sialyl-Tn (STn) (Neu5Acα2,6GalNAcα1-Ser/Thr) [83,150]. Tumor-associated changes in glycosylation drive cancer progression and metastasis, providing novel diagnostic biomarkers and therapeutic targets [83]. Current applications of sophisticated mass spectrometry approaches in clinical diagnosis coupled with the advances in exome sequencing in patients have contributed to boosting the number of new glycosylation disorders [151]. 

In this review, we have illustrated how the steps of glycan modifications are intimately connected to secretory trafficking with perturbation in Golgi glycosylation resulting from mutations in proteins required for COPI coat scaffolding, the incorporation of glycosyltransferases in vesicles and vesicle tethering. In the context of *N*-glycosylation GORAB, GOLPH3 and the COG complex are all required for the normal synthesis of complex glycans, with effects on sialylation of *N-*glycans. However, these proteins act at different steps of this process, with GORAB acting as a scaffold for the COPI coat complex, GOLPH3 regulating the incorporation of a subset group of glycosyltransferases into COPI vesicles, and the COG complex mostly being involved in vesicle tethering. However further studies should clarify the reciprocal dependence between these vesicle trafficking players in *N-* and *O-*glycosylation, and whether they display a specificity towards defined groups of glycosyltransferases and a cell-type limited function. Although the discovery of novel glycosylation disorders is increasing rapidly, genetic variants of important trafficking regulators have not been studied within the context of CDGs to date. For example, GOLPH3 and COG3 have been shown to have a role in glycosylation in model organisms and mammalian cell cultures, but they have not yet been associated with a CDG [65,76,81,106]. We may expect that the potential impact of mutations in some secretory trafficking players on glycosylation is so severe that it is not compatible with viability. Studies on model organisms (e.g., zebrafish, flies, mice) provide a valuable resource with which to dissect the phenotypic consequences of glycosylation disfunction in the whole organism and to identify potential therapeutic strategies for CDGs or cancer aimed at modulating secretory trafficking pathways [152].

## Figures and Tables

**Figure 1 cells-09-02652-f001:**
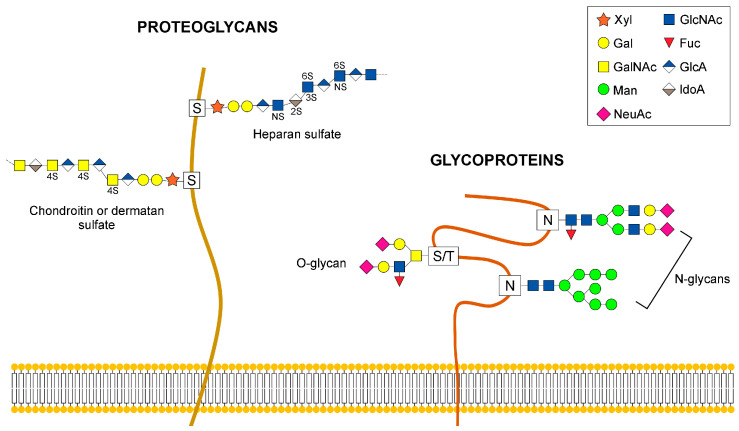
Examples of glycans that are elaborated in the Golgi. The diagram depicts *N-* and *O-*linked glycans attached to glycoproteins and proteoglycans. Abbreviations are: Xyl, xylose; Gal, galactose; GalNAc, *N*-acetylgalactosamine; Man, mannose; NeuAc, *N*-Acetylneuraminic acid; GlcNAc, *N*-acetylglucosamine; Fuc, Fucose; GlcA, glucuronic acid; IdoA, iduronic acid.

**Figure 2 cells-09-02652-f002:**
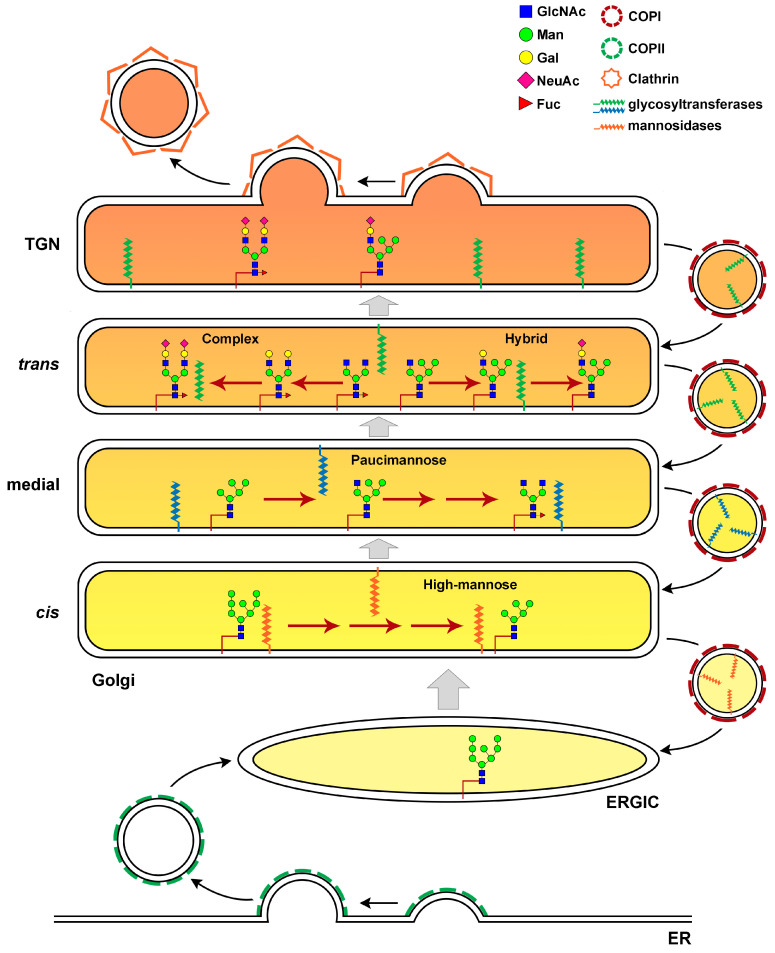
Elaboration of *N-*linked Glycans in the Golgi cisternae. *N-*glycan processing in the Golgi gives rise to three main classes of glycans: high-mannose, hybrid and complex glycans. Glycoproteins exit the ERGIC and reach the Golgi apparatus with eight or nine mannose residues. The Golgi apparatus is organized into discrete cisternae where the enzymes needed for glycan elaboration are specifically compartmentalized. Mannose trimming enzymes are located in the *cis* and *medial*-Golgi. The addition of GlcNAc is an essential step to generate hybrid and complex *N-*linked glycans in the *trans*-Golgi. Hybrid *N-*glycans are generated by the extension of the α3-arm that received GlcNAc with the addition of Gal, Fuc and SA. Removal of two terminal of the five Man residues, allows branching with an additional GlcNAc residue and synthesis of biantennary complex *N-*glycans with Gal, Fuc and SA. The polarized distribution of glycosylation enzymes is maintained through retrograde transport of COPI coated vesicles. A proper cisternal pH (different shades of orange) is needed to control retrograde sorting of enzymes and the association of glycoenzymes with the trafficking cargo. GlcNAc, *N*-acetylglucosamine; Man, mannose; Gal, galactose; NeuAc, *N*-Acetylneuraminic acid; Fuc, Fucose.

**Figure 3 cells-09-02652-f003:**
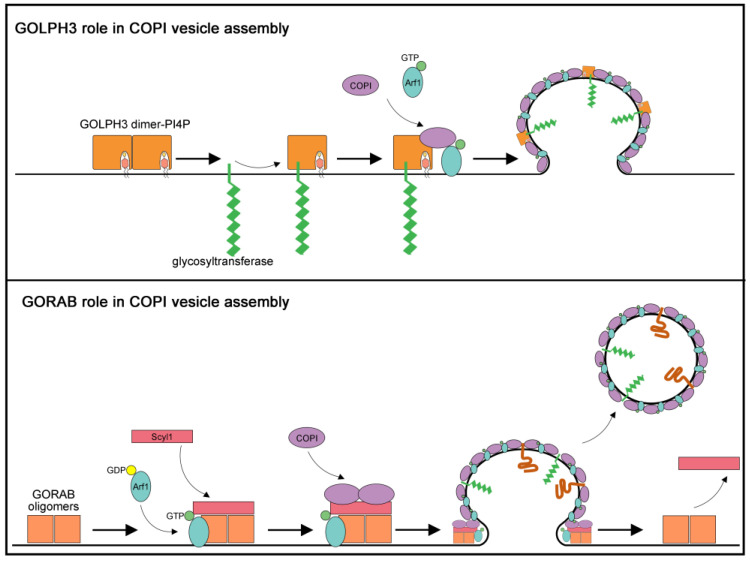
Proposed models for roles of GORAB and GOLPH3 in COPI-mediated trafficking. (Upper panel) The diagram depicts a model for GOLPH3 role in *N-*glycosylation based on the results of Eckert and coauthors [65]. GOLPH3 protein binds to the coatomer as well as to a set of glycosyltransferases, such as C2GnT and SiaT as well as the coatomer, thereby mediating incorporation of these enzymes into COPI coated vesicles. (Lower panel) The diagram depicts a model for GORAB role in COPI vesicle release based on the results of Witkos and coauthors [66] GORAB self-associates in oligomers that form discrete domains at the *trans-*Golgi membrane. GORAB oligomers recruit Scyl1 protein driving Arf1-GTP accumulation in the GORAB-enriched domains. High levels of Scyl1 and Arf1-GTP proteins lead to the selective COPI loading. In turn, coatomer assembly leads to vesicle budding and cargo incorporation while GORAB/Scyl1 complex stabilizes the bud neck during vesicle formation. After releasing of the COPI vesicle from the Golgi membrane, GORAB and Scyl1 detach from the complex to start a new cycle.

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
