# Peer review of "The Close Relationship between the Golgi Trafficking Machinery and Protein Glycosylation"

_cells, 2020, doi:10.3390/cells9122652_

Round 1

Reviewer 1 Report

The manuscript by Frappaolo et al. provides an interesting review of glycosylation in the Golgi and its relationship to proteins involved in trafficking. The review focuses on the GOLPH3 and GORAB proteins, and evidence for their roles linking these processes. One concern is that the authors seem not to take a critical eye to the literature, and do not distinguish between high quality and low quality published data (especially where it supports their model). However, this manuscript is a Review, and the authors deserve leeway to present their view of the literature. As such, I think this review as written serves its purpose.

Author Response

We think that all the papers that we cited demonstrate the link between Golgi trafficking regulators and protein glycosylation. The aim of our review is to report an overview of the roles of GOLPH3, GORAB and the COG complex in glycosylation. Our review suggests that these proteins act at different steps of glycosylation with GORAB acting as a scaffold for the COPI coat complex, GOLPH3 regulating the incorporation of a subset group of glycosyltransferases into COPI vesicles and the COG complex mostly involved in vesicle tethering. As reported in the conclusions further studies should clarify the reciprocal dependence between these vesicle trafficking players in N- and O-glycosylation and whether they display a specificity towards defined groups of glycosyltransferases and a cell-type limited function.

Reviewer 2 Report

This review from Frappaolo et al. describes the relationship between the Golgi trafficking machinery, glycosyltransferases and protein glycosylation. Overall, the review is well written and I have only a few comments that should be addressed by the reviewers.

Major comments:

Most of the review covers published work on different Golgi trafficking machineries, which is only to some extent new. I am missing a more general discussion or mechanistic model how the different Golgi adaptor proteins/trafficking complexes regulate glycosyltransferase localization and protein glycosylation. For example, what do the GOLPH3 interacting glycosyltransferases acting in different glycosylation pathways have in common (apart from the interacting motif)? Do GORAB, GOLPH3 and the COG complex regulate different Golgi trafficking steps of the same glycosyltransferase (e.g. ST6Gal) or are they involved in cell-type specific processes or glycosyltransferase/glycosylation-specific processes?

Minor comments:

Line 22: for N-glycosylation, protein glycosylation is also co-translational, please add this to the posttranslational modification.

Line 35: N-glycolylneuraminic acid instead of N-acetylglycolylneuraminic acid.

Line 42: folding processes instead of folding process.

Lines 75-76: In the cisternal maturation model – which is favoured by the authors – cargo transport is not regulated by vesicle trafficking. Please check and modify the sentence accordingly.

Line 87: add also more recent Refs related to the cisternal maturation model. Mention at least that there are also other models including for example tubular connections between cisternae.

Figure 2: the mannosidase/glycosidase shown in red in the cis Golgi is not mentioned in description of the symbols.

Lines 254 and 300: isn’t this the same enzyme? If yes, please make it clear and use the same abbreviation.

Author Response

Major comments:

Most of the review covers published work on different Golgi trafficking machineries, which is only to some extent new. I am missing a more general discussion or mechanistic model how the different Golgi adaptor proteins/trafficking complexes regulate glycosyltransferase localization and protein glycosylation. For example, what do the GOLPH3 interacting glycosyltransferases acting in different glycosylation pathways have in common (apart from the interacting motif)? Do GORAB, GOLPH3 and the COG complex regulate different Golgi trafficking steps of the same glycosyltransferase (e.g. ST6Gal) or are they involved in cell-type specific processes or glycosyltransferase/glycosylation-specific processes?

To meet the reviewer’s suggestion we amended the Concluding remarks. Specifically we added the following text: “In the context of N-glycosylation GORAB, GOLPH3 and the COG complex are all required for normal synthesis of complex glycans, with effects on N-sialylation. However, that these proteins act at different steps of glycosylation with GORAB acting as a scaffold for the COPI coat complex, GOLPH3 regulating the incorporation of a subset group of glycosyltransferases into COPI vesicles and the COG complex mostly involved in vesicle tethering. However further studies should clarify the reciprocal dependence between these vesicle trafficking players in N- and O-glycosylation and whether they display a specificity towards defined groups of glycosyltransferases and a cell-type limited function.”

Minor comments:

Line 22: for N-glycosylation, protein glycosylation is also co-translational, please add this to the posttranslational modification.

We modified the text as suggested. See line 23.

Line 35: N-glycolylneuraminic acid instead of N-acetylglycolylneuraminic acid.

We modified the text as suggested. See line 35.

Line 42: folding processes instead of folding process.

We modified the text as suggested. See line 43.

Lines 75-76: In the cisternal maturation model – which is favoured by the authors – cargo transport is not regulated by vesicle trafficking. Please check and modify the sentence accordingly.

We modified the sentence as suggested. See line 74-75.

Line 87: add also more recent Refs related to the cisternal maturation model. Mention at least that there are also other models including for example tubular connections between cisternae.

To meet the reviewer’s suggestions we cited two more recent References related to the cisternal maturation model. Moreover we changed the text (Lines 105-110) to cite the tubular connections between cisternae.

Figure 2: the mannosidase/glycosidase shown in red in the cis Golgi is not mentioned in description of the symbols.

We modified Figure 2 to include the description of the mannosydase/glycosydase shown in red.

Lines 254 and 300: isn’t this the same enzyme? If yes, please make it clear and use the same abbreviation.

We used the same abbreviation to indicate  a-2,6-sialyltransferase: ST6GAL1.